# Farmland Carbon and Water Exchange and Its Response to Environmental Factors in Arid Northwest China

Xinqian Zheng [1], Fan Yang [2,3,4,*], Ali Mamtimin [2,3,4], Xunguo Huo [1], Jiacheng Gao [2,3,4], Chunrong Ji [1], Silalan Abudukade [2,3,4], Chaofan Li [5], Yingwei Sun [6], Wenbiao Wang [7], Zhengnan Cui [7], Yu Wang [2,3,4], Mingjie Ma [2,3,4], Wen Huo [2,3,4], Chenglong Zhou [2,3,4] and Xinghua Yang [2,3,4]

[1] Xinjiang Agro-Meteorological Observatory, Xinjiang University, Urumqi 830002, China; zhengxinqian2018@yeah.net (X.Z.); huoxunguo2018@yeah.net (X.H.); jichr@idm.cn (C.J.)
[2] Institute of Desert Meteorology, China Meteorological Administration, Urumqi 830002, China; ali@idm.cn (A.M.); gaojiach@idm.cn (J.G.); 107552101047@stu.xju.edu.cn (S.A.); wangyu@idm.cn (Y.W.); mamj@idm.cn (M.M.); huowenpet@idm.cn (W.H.); zhoucl@idm.cn (C.Z.); yangxh@idm.cn (X.Y.)
[3] National Observation and Research Station of Desert Meteorology, Taklimakan Desert of Xinjiang, Urumqi 830002, China
[4] Taklimakan Desert Meteorology Field Experiment Station, China Meteorological Administration, Urumqi 830002, China
[5] Collaborative Innovation Center on Forecast and Evaluation of Meteorological Disaster, School of Geographic Sciences, Nanjing University of Information Science and Technology, Nanjing 210044, China; lcf@nuist.edu.cn
[6] Xinjiang Information Engineering School, Xinjiang University, Urumqi 830002, China; yingwei_sun@yeah.net
[7] Elion Resources Group Co., Ltd., NO.15 Guanghua Road, Chaoyang District, Beijing 100026, China; wangwenbiao@elion.com.cn (W.W.); cuizhengnan@elion.com.cn (Z.C.)
* Correspondence: yangfan309@yeah.net; Tel.: +86-(0991)-2652429

**Abstract:** Carbon neutrality is an important target in China's efforts to combat the climate crisis. The implementation of carbon neutrality requires high crop yields in farmland ecosystems of arid regions. However, the responses of farmland ecosystems to environmental changes and their effects on the conversion and intensity of carbon sources/sinks within farmlands in arid regions remain unclear, which limits carbon sequestration. In this study, we used a set of eddy covariance systems to observe carbon and water fluxes in cotton and spring maize, two typical crops in arid regions of Northern Xinjiang in China. The carbon and water exchange and water use efficiency (WUE) of cotton and spring maize were evaluated over the entire growth cycle with respect to changes in the environment. Our results show that the carbon sequestration capacity of farmland ecosystems in arid regions is undeniable and is strongly influenced by the growth and development of plants. Spring maize, as a representative of C4 plants, exhibited a 58.4% higher carbon sequestration efficiency than cotton, a C3 plant, and they both reached their carbon sequestration efficiency peak in July. Throughout the growth period, temperature, net surface radiation, and saturated vapor pressure differences (VPD) significantly affected the carbon sequestration capacity and WUE of both crops. Optimal temperatures can maximize the carbon sequestration efficiency of cotton and spring maize; for cotton, they are 20–25 °C, and for spring maize, they are 22–27 °C, respectively. In addition, it is recommended that spring maize be harvested at the end of July when it meets the harvesting standards for silage feed and achieves the maximum carbon sequestration. Afterward, winter crops should be planted to maximize the yield and improve the carbon sequestration capacity of farmlands.

**Keywords:** $CO_2$ flux; cotton field; maize field; eddy covariance; water use efficiency

## 1. Introduction

The terrestrial ecosystem is an important part of the Earth's carbon pool, and small changes in the terrestrial ecosystem can have a profound impact on the total amount of atmospheric $CO_2$ and promote the positive feedback effects of climate change [1–4]. Thus, carbon sequestration can be more effective by enhancing, restoring, and optimizing the

structure, function, and spatial layout of the terrestrial ecosystem. This is an important path for achieving the goals of carbon peaking and carbon neutrality, and it also provides the most economical, effective, and safest means of promoting carbon sequestration [5]. According to the Food and Agriculture Organization of the United Nations, the total agricultural land in the world is about 4.7 billion hectares, i.e., approximately one-third of the global land area [6]. Greenhouse gas emissions from agricultural land exceed 30% of the global total anthropogenic greenhouse gas emissions, which are equivalent to generating 15 billion tons of carbon dioxide annually. As a major agricultural country, China's "14th Five-Year Plan for National Green Agriculture Development" released in 2021 has made carbon reduction and sequestration in agriculture and rural areas important goals. By monitoring carbon flux in farmlands to ensure and improve crop yield, a deep understanding of the carbon budget of farmland ecosystems and its relationship with environmental and human management factors will allow the implementation of appropriate agricultural management measures to further reduce carbon emissions and increase carbon storage in farmland ecosystems. This is of great significance for achieving the goals of "carbon peaking" and "carbon neutrality" as soon as possible.

In order to accurately evaluate the land–atmospheric exchange process of water vapor and $CO_2$ fluxes, various research methods have been developed on different research scales, including the open-top growth chamber (OTC) [7], Burnby energy balance method [8], eddy covariance method (EC) [9], and large aperture scintillator method (LAS) [10]. Among these methods, the eddy covariance method (EC) is considered to be one of the most direct and reliable methods for obtaining data on $CO_2$, water vapor, and heat fluxes for a soil–vegetation–atmosphere system, as the EC method has a solid theoretical basis, high observation accuracy, and continuous stability [11–13]. However, the results obtained using EC methods can only represent the characteristics of a specific ecosystem in a specific environment. To extrapolate research results to regional or global scales, they must be combined with relevant models to decipher complex interactions between terrestrial carbon and water dynamics and global changes, as well as for quantitative simulation and prediction of carbon and water dynamics. Models widely used at present for this purpose include SIB2 [14,15], CEVSA [16], BIOME-BGC [17], Crop-C [18], DNDC [19], CENTRY [20], ORCHIDE-STICS [21], JULES CROP [22], etc.

The farmland ecosystem is an ecosystem where human activities are most frequent and intense. The carbon cycle in farmland ecosystems is mainly involved in the exchange process between the plant and soil carbon pools and the external environment, as well as in the migration and transformation of different components of a carbon pool [23]. Due to significant differences in carbon fluxes of different crops and regions, farmland ecosystems may serve as both carbon sources and carbon sinks. Using the eddy covariance method, researchers have determined that the carbon flux of farmland ecosystems in the growth season exhibits a U-shaped curve [24–27]. In particular, farmlands release $CO_2$ at night; after sunrise, farmlands change from carbon sources to carbon sinks, reaching a daily carbon sequestration peak around noon [28,29]. Some crops also experienced a "midday depression of photosynthesis" phenomenon at noon, with a daily carbon flux curve in the shape of a "W" [30]. The growth of crops is comprehensively influenced by various factors, such as photosynthetic active radiation, temperature, precipitation, soil moisture, plant development, and leaf area index. These factors affect the carbon sequestration capacity of farmland ecosystems, which is significantly lower in the early and late stages of crop growth than in the period of high yield [31,32]. The main influencing factors of the same crop vary with different time scales. For example, the carbon flux of maize is most affected by the net radiation on the hour scale. At the daily scale, the spring maize is most influenced by the saturation vapor pressure difference, while the summer maize is most affected by temperature [30]. Studies have also indicated that the leaf area index, which reflects crop growth, is the main controlling factor of daily carbon flux in crops. Additionally, net radiation is the primary controlling factor in crop evapotranspiration on a daily scale [33]. Under the multi-cropping mode, the carbon flux of farmlands roughly

shows a "W" shaped bimodal curve [34,35]. Different field management measures also have an impact on farmland carbon flux. For instance, the narrow row spacing is conducive to the formation of a high biomass and leaf area index, thereby promoting high carbon sequestration [36]. Compared to traditional tillage methods, the net carbon exchange under no tillage is neutral [37].

Xinjiang in northwest China encompasses both arid and semi-arid regions; it is among the most sensitive regions to global change worldwide. It has 7066 thousand hectares of cultivated land, of which 96% is irrigated. Efficient water use means stronger carbon sink capacity and higher crop yield [38]. This is of great significance for estimating farmland yield, efficient water resource utilization, and carbon sequestration and sink enhancement in arid regions. Relying on its abundant light and heat along with vast land, Xinjiang has become the largest production base of high-quality commodity cotton in China. In 2021, the total cotton planting area in Xinjiang was $2.48 \times 10^4$ km$^2$, with a total output of 5.129 million tons, accounting for 89.5% of the total cotton output in China. The cotton industry has become a pillar of Xinjiang's economy and agricultural development. In 2021, the planting area of maize in Xinjiang was $1.11 \times 10^4$ km$^2$, with a total output of 10.1265 million tons [39]. Maize is expected to play an increasingly important role in the national economy of Xinjiang.

With the increasingly severe impact of climate change on ecosystem stability, the implementation of the carbon-neutral target requires farmland ecosystems to ensure high crop yields. For arid regions, where water resources are extremely scarce, it is more challenging to achieve high crop yields. Therefore, we chose two typical agricultural fields (cotton and spring maize) in arid regions of northern Xinjiang, China, as the focus of our research. In 2018, we installed an eddy covariance system at the interface between adjacent cotton and spring maize fields and segmented the observed $CO_2$ and $H_2O$ turbulent flux based on the flux source partitioning analysis. We used this system to compare and analyze the differences in carbon and water exchange characteristics, water use efficiency (WUE), and their relationship with environmental factors in cotton and spring maize fields. The purpose of this study is to understand the carbon and water exchange characteristics and differences of typical crops in arid areas and fully utilize the efficiency of instrument observation. At the same time, this study provides basic support for the accurate assessment of the carbon sequestration capacity of farmland in arid regions, agricultural policy formulation, rational allocation of agricultural resources, and implementation of appropriate farmland management measures.

## 2. Materials and Methods

### 2.1. Site Description

The research area is located in Paotai Town, Shihezi City, Xinjiang (44°49′ N, 85°33′ E, elevation: 300 m a.s.l.), situated at the northern central foot of the Tianshan Mountains and the southwest bottom of the Junggar Basin, within Shawan County in the middle section of the Tianshan North Slope Economic Belt. It has a temperate arid continental climate, with hot summers and cold winters. The annual average temperature is 7.9 °C, with a sunlight duration of 2771.3 h and an average annual precipitation amount of 162 mm. The annual average evaporation is 1892.8 mm, which is more than 10 times of precipitation. And the annual average frost-free period is 190 days. There are mostly gray desert soil, tidal soil, and meadow soil that are suitable for the growth of various plants. Irrigation agriculture is the main industry in this area, with major crops like cotton, maize, wheat, grapes, tomatoes, watermelons, etc.

As shown in Figure 1, during the experimental observation period (May to October 2018), the daily average temperature ($T_a$) was 20.2 °C, ranging from 2.9 °C to 31.8 °C. The highest temperature occurred on 11 August 2018, and the lowest on 30 October 2018. The daily average temperature of June was the highest at 25.7 °C, followed by July and August at 25.1 °C and 24.9 °C, respectively. The prevailing wind directions were east/northeast and west/southwest, with a daily average wind speed of 1.36 m·s$^{-1}$. More than 80% of days

had wind speeds below 2 m·s$^{-1}$. The maximum wind speed was 7.83 m·s$^{-1}$, occurring on 24 May 2018. The daily average saturation vapor pressure difference (VPD) was 1.15 KPa, with a range of 0.147–3.196 KPa. The daily average net radiation was 130.6 W·m$^{-2}$, and the overall variation trend was similar to that of temperature.

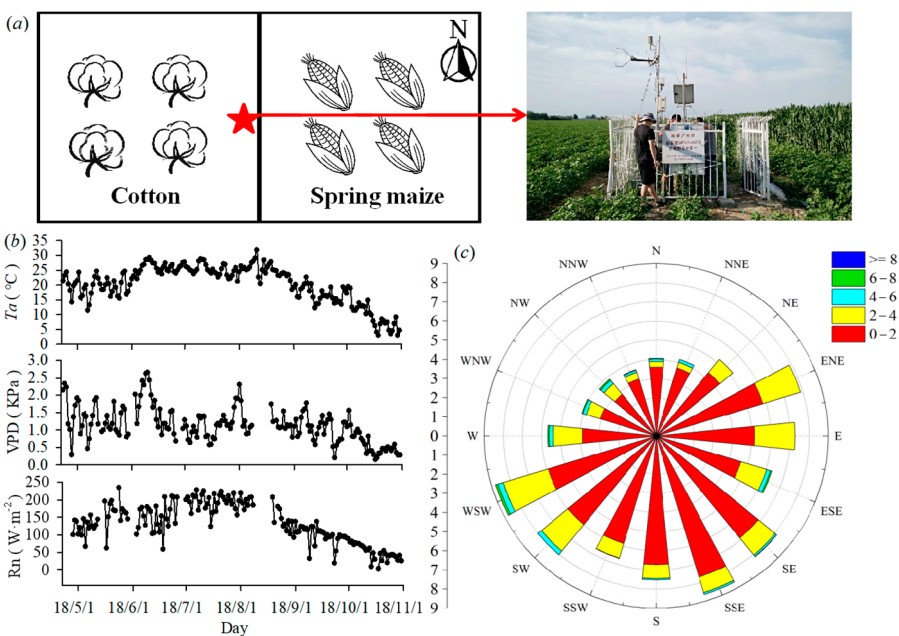

**Figure 1.** (**a**) Observation layout and location of observation stations in the study area (red star). (**b**) The temporal variation characteristics of the environmental factors: $T_a$, air temperature; VPD, saturated vapor pressure difference; Rn, net radiation. (**c**) Wind rose: wind speed frequency distribution in different wind directions.

## 2.2. Experimental Design

The eddy covariance system was set up at the interface of adjacent fields planted with cotton and spring maize. All micrometeorological observations were conducted from May to October 2018 (Figure 1). Cotton was planted on the west side, and spring maize was planted on the east side. The observed $CO_2$ flux was split according to the flux source area. There were 3632 westerly data and 4099 easterly data, with a ratio of 0.89:1. The westerly turbulent flux represented cotton, and the easterly turbulent flux represented spring maize. Cotton was sown on 16 April 2018, emerged on 30 April, and stopped growing on 8 October. It was harvested on 25 October. Spring maize was sown on 8 April 2018, sprouted on 20 April, and matured on 14 August. It was harvested on 27 August. The main growth stages and characteristics are shown in Table 1, and the two crops both used drip irrigation under film.

**Table 1.** Main development stages and basic situation of cotton and spring maize.

| Cotton | | | | Spring Maize | | | |
|---|---|---|---|---|---|---|---|
| Growth Period | Main Period | Plant Height (cm) | Irrigation Volume (m³·hm⁻²) | Growth Period | Main Period | Plant Height (cm) | Irrigation Volume (m³·hm⁻²) |
| Sowing | Mid-April ~ Late April | - | 675 | Sowing | Early April ~ Mid-April | - | 600 |
| Seedling | Early May ~ Early June | 28 | - | Seedling | Late April ~ Early May | 36 | - |
| Budding | Mid-June ~ Late June | 50 | 1350 | Jointing | Late May ~ Mid-June | 106 | 1550 |
| Flowering and boll | Early July ~ Late Aug | 63 | - | Tasseing stage | Late June ~ Mid-July | 221 | 1700 |
| Boll opening | Early Sept ~ Late Sept | 66 | 2100 | Milky maturation stage | Late July ~ Early Aug | 230 | 1850 |
| | | | | Maturity | Mid-Aug ~ Late Aug | 230 | - |

The eddy covariance system was installed 3.0 m above ground level, and its height remained fixed throughout the entire growth period of plants. The system included a three-dimensional ultrasonic anemometer and an open-path infrared gas analyzer. The data

were measured at a frequency of 10 Hz and stored in a data collector (CR3000 Micrologger, Campbell Scientific, Logan, UT, USA). The $CO_2/H_2O$ flux ($F_C/F_H$) was calculated using the covariance of the fluctuating values of vertical wind speed ($w'$) and $CO_2/H_2O$ concentration ($\rho'$). We used a four–component radiometer installed on the mounting arm (due south orientation) at a height of 1.5 m to observe upward/downward short-wave and long-wave radiation. In addition, two soil temperature sensors were buried 0 cm and 5 cm underground, and a soil humidity sensor and a soil heat flux plate were buried 5 cm underground near the eddy covariance system. Table 2 provides a detailed description of the instruments used in this experiment.

**Table 2.** The basic information of experimental observation systems.

| Observation Systems | Sensors | Observation Variables | Height/Depth (m) | Acquisition Frequency |
|---|---|---|---|---|
| The eddy covariance system | Three–dimensional ultrasonic anemometer (CSAT3, Campbell Scientific Inc., Logan, UT, USA) Open–path infrared gas analyzer (LI-7500A, LI-COR Inc., Lincoln, NE, USA) | Three-dimensional wind speed and direction, $T_a$, $RH$, $CO_2/H_2O$ concentration, H, LE, $Fc$, $F_H$ | 3.0 | 10 Hz |
| The environmental element observation system | Four-component radiometer (CNR4, Kipp and Zonen B. V., Delft, The Netherlands) Soil temperature sensors (109, Campbell Scientific Inc., UT, USA) Soil humidity sensor (Hydra 93640, Stevens Inc., Portland, OR, USA) Soil heat flux plate (HFP01SC, Hukseflux, Inc., Delft, The Netherlands) | DR, UR, DLR, ULR, Rn | 1.5 | 1 Hz |
| | | Soil temperature | 0, $-0.05$ | |
| | | Soil moisture | $-0.05$ | |
| | | Soil heat flux | $-0.05$ | |

where $T_a$, $RH$, H, LE, $F_C$, $F_H$, DR, UR, DLR, ULR, and Rn were air temperature, relative humidity, sensible heat flux, latent heat flux, $CO_2$ flux, $H_2O$ flux, total solar radiation, ground-reflected radiation, atmospheric long-wave radiation, ground long-wave radiation, and net radiation, respectively.

### 2.3. Data Processing

Conventional Meteorological Data

EddyPro Software v7.0.9 was used to process the original data. Data processing included the detection and removal of peaks [40], sonic temperature correction [41], two-dimensional coordinate rotation [42], frequency response correction [43], and Webb–Pearman–Leuning density correction [44]. After processing, data collected during rainfall were eliminated. Finally, the fluxes of sensible heat (H), latent heat (LE), and $CO_2/H_2O$ ($F_C/F_H$) were obtained every 30 min. These values were calculated as follows:

$$H = \rho C_\rho \overline{w'T'} \tag{1}$$

$$\text{LE} = L_V \overline{w'\rho'_V} \tag{2}$$

$$F_C = \overline{w'\rho'_C} \tag{3}$$

$$F_H = \overline{w'\rho'_V} \tag{4}$$

where $\rho$ is the air density (kg·m$^{-3}$), $C_\rho$ is the specific heat of air at a constant pressure, w' is the fluctuating value of the vertical wind speed, $T'$ is the potential temperature fluctuation value, $\rho'_V$ and $\rho'_C$ are the fluctuating values of water vapor concentration and $CO_2$ concentration, respectively, and $L_V$ is the latent heat of evaporation.

### 2.4. Statistical Analyses

VPD can be estimated from the relative humidity ($RH$) of the air temperature ($T_a$) as follows:

$$VPD = 0.611 \times e^{\frac{17.27 \times T_a}{T_a + 237.3}} \times (1 - \frac{RH}{100}) \tag{5}$$

WUE$_e$ is the ecosystem-level WUE. It is the ratio of the yield (kg·hm$^{-2}$) to the total water (rainfall + irrigation, m$^3$·hm$^{-2}$). WUE$_l$ is the leaf-level WUE. It is the ratio of the net ecosystem carbon exchange (*NEE*, mg·m$^{-2}$·s$^{-1}$) to the evapotranspiration (*ET*, mm·s$^{-1}$).

SigmaPlot 14.0 and Origin Pro 9.0 were used for data calculation and statistical analysis. Significance was evaluated at $p < 0.05$ or $p < 0.01$, as indicated. We also analyzed the correlation between $F_C$ and WUE with various environmental parameters to identify key influencing factors.

## 3. Results

### 3.1. Monthly Mean Diurnal Variation of Sensible and Latent Heat Fluxes

Terrestrial ecosystems exchange heat and water vapor with the atmosphere in the form of H and LE. As shown in Figure 2, the diurnal variation in H during the cotton and spring maize growth and development stages was generally consistent, with a basic distribution of a single peak that first rises and then falls. H was slightly lower for cotton than for spring maize in all stages, and the peak occurred earlier in cotton than in spring maize. At different stages, both cotton and spring maize exhibited a high-low-high trend in H throughout the growing season. The peak value of H in cotton occurred in May (169.3 W·m$^{-2}$), followed by September (155.2 W·m$^{-2}$) and the lowest value occurred in July (−4.3 W·m$^{-2}$). The peak value of H in spring maize took place in September (225.2 W·m$^{-2}$), followed by May (163.5 W·m$^{-2}$), with the lowest value occurring in July (13.9 W·m$^{-2}$).

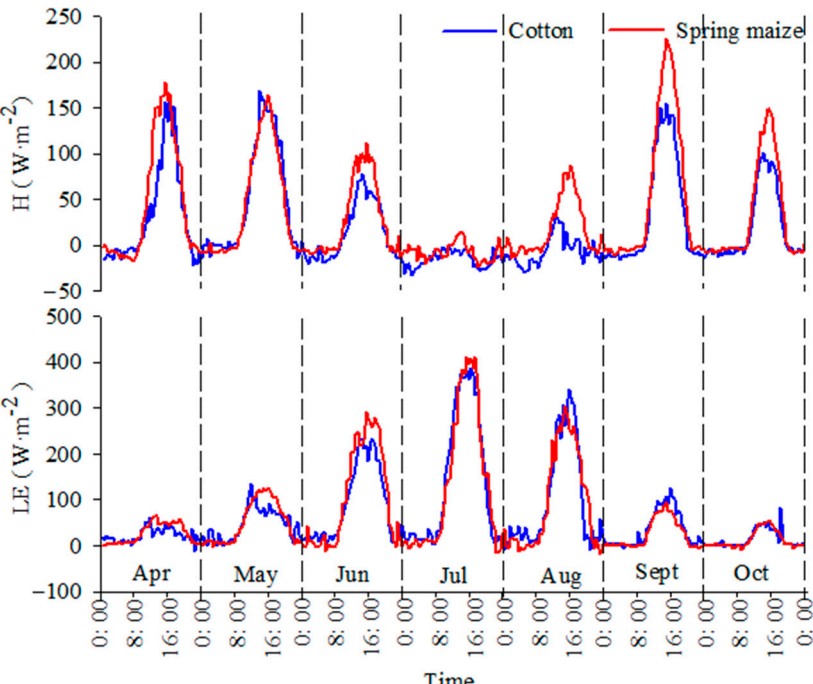

**Figure 2.** Monthly average daily variation in sensible heat flux (H) and latent heat flux (LE) of cotton and spring maize.

Diurnal variation in latent heat flux at different growth stages of cotton and spring maize was similar to those of H, with a single-peak distribution with an initial rise and then a fall. At different stages, both cotton and spring maize exhibited a low-high-low trend in LE throughout the growing season. In general, the LE of cotton was slightly lower than that of spring maize at all stages, and the timing of their respective peak values was relatively consistent, with both occurring in July (cotton, 383.3 W·m$^{-2}$; spring maize, 410.6 W·m$^{-2}$), followed by August (cotton, 339.9 W·m$^{-2}$, spring maize, 304.1 W·m$^{-2}$), and June (cotton, 233.2 W·m$^{-2}$, spring maize, 292.9 W·m$^{-2}$). Overall, under the combined influence of surface energy and plant growth, there was a significant complementary effect between H and LE of spring maize and cotton during the growth period. And the complementary

effect was most prominent in July when relatively more energy was transferred from the ground in the form of LE due to the comparatively lower H.

### 3.2. Monthly Mean Diurnal Variation of $CO_2$ Flux and Its Correlation with Meteorological Factors

The $F_C$ of farmland ecosystems was mainly affected by the combined effects of crop photosynthesis, respiration, and soil microbial respiration. The $F_C$ of cotton and spring maize exhibited an apparent single-peak U-shaped distribution during the main growing season (June–August), as shown in Figure 3. At around 10:00 (after sunrise), photosynthesis began to increase, reaching its peak at around 14:00. Then, it began to decline, becoming negative during the day, which indicates that the ecosystem was a carbon sink. Before sunset, photosynthesis gradually weakened until 20:00, and the respiration of crops and soil microorganisms released $CO_2$. Consequently, $CO_2$ concentrations within the canopy increased. $F_C$ became positive overnight, which indicates that the ecosystem was a carbon source. This phenomenon was most evident in July, with the peak absorption rate of cotton reaching 1.023 mg·m$^{-2}$·s$^{-1}$ and that of spring maize reaching 1.312 mg·m$^{-2}$·s$^{-1}$. Notably, cotton, as a C3 plant, has lower photosynthesis efficiency than C4 plants like spring maize. Therefore, the carbon sequestration rate of cotton was lower than that of spring maize during the main growing season in our study. Additionally, the aboveground stems of spring maize grew faster and exhibited daytime $CO_2$ absorption features as early as May, while cotton showed more obvious $CO_2$ absorption characteristics only in June. As spring maize was harvested at the end of August, its carbon sequestration rate was slightly lower than that of cotton.

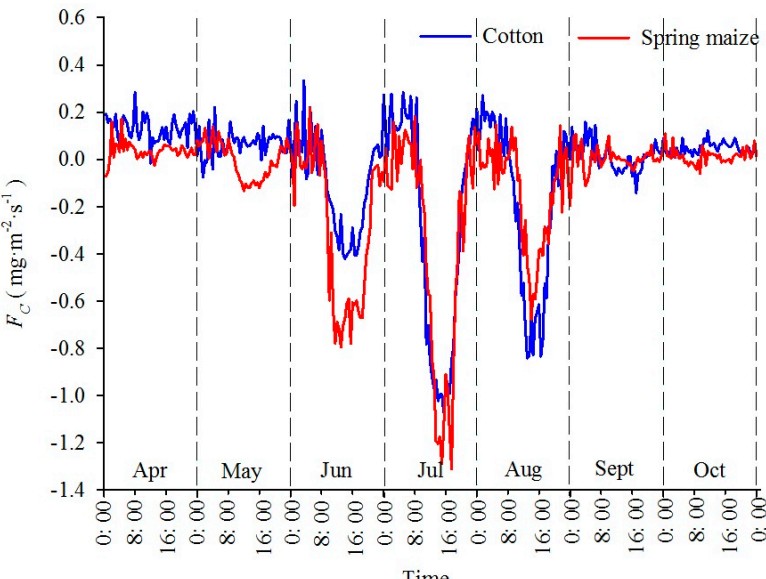

**Figure 3.** Monthly average daily variation in $CO_2$ flux ($F_C$) for cotton and spring maize.

Data from months with large fluctuations in $CO_2$ flux (May–September) during the daytime (10:00–20:00) were selected for correlation analysis with $T_a$, VPD, and net radiation (Rn) (Table 3). The results showed that solar radiation was the main factor controlling the $CO_2$ absorption capacity of ecosystems, followed by temperature. Notably, the correlation between the $CO_2$ absorption capacity of ecosystems and VPD was weak. As Xinjiang is an irrigated agricultural region, water conditions do not limit the growth of crops. Fitting the nonlinear relationship between $CO_2$ flux, Rn, and temperature (Figure 4) revealed that as the temperature and solar radiation increased, $CO_2$ absorption gradually strengthened, particularly during high solar radiation stages when absorption was more pronounced. This finding indicates that increased solar radiation and temperature significantly promote the carbon sequestration rate. However, the $CO_2$ flux intensity decreased when the temperature was too high. Besides, the optimal temperature for cotton growth ranged from

20 °C to 25 °C, at which cotton had its highest net photosynthetic production rate and fastest growth, as it is a state of thermal saturation. At temperatures above 28 °C, the photosynthetic production rate of the cotton crop decreased, and plant growth slowed down significantly. At temperatures below 20 °C, heat was insufficient for optimal cotton growth. The optimal temperature for spring maize growth ranged from 22 °C to 27 °C, at which the net photosynthetic production rate was highest and growth was fastest. However, the photosynthetic production rate decreased at temperatures below 20 °C or above 30 °C.

**Table 3.** Correlation coefficient (R) values for the relationships between $CO_2$ flux ($F_C$) and environmental factors in cotton and spring maize.

|  | $T_a$ | VPD | Rn |
|---|---|---|---|
| $F_C$ of Cotton | −0.387 * | 0.083 | −0.592 ** |
| $F_C$ of Spring maize | −0.417 ** | −0.127 | −0.456 ** |

$T_a$: daily average air temperature; VPD: saturated vapor pressure difference; Rn: net radiation. * $p < 0.05$; ** $p < 0.01$.

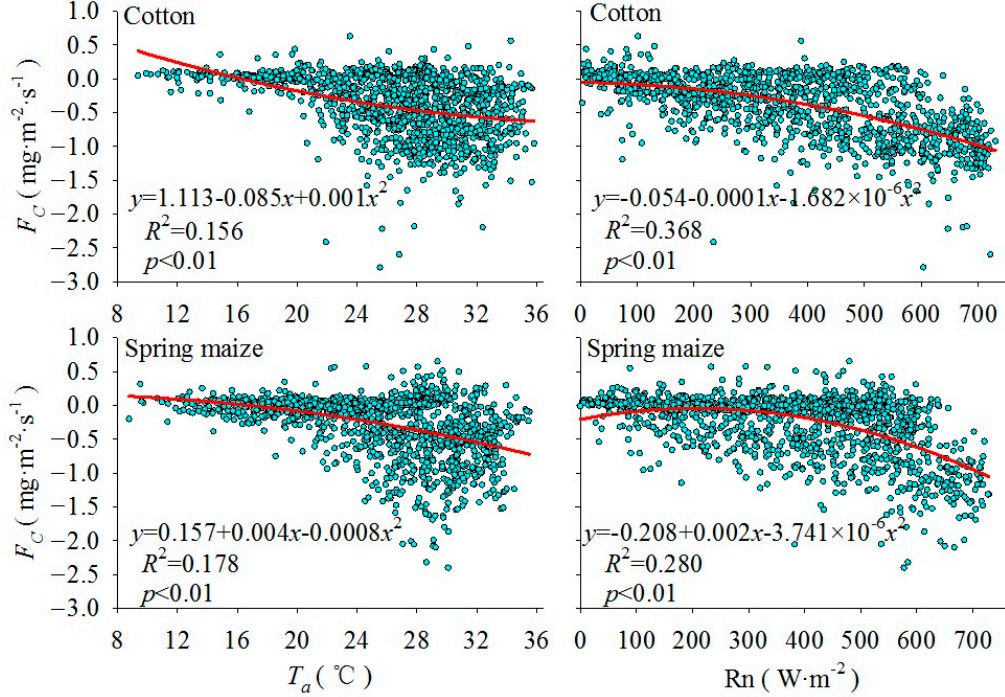

**Figure 4.** Relationships between $F_C$ and daily average temperature ($T_a$) and net radiation (Rn) in cotton (**upper**) and spring maize (**lower**).

*3.3. Monthly Average Daily Variation of Water Vapor Flux and Its Correlation with Meteorological Factors*

The variation features of water vapor flux in cotton and spring maize were consistent with the corresponding latent heat flux. The peak values of water vapor flux for each month ranged between 21.095 mg·m$^{-2}$·s$^{-1}$ and 152.337 mg·m$^{-2}$·s$^{-1}$ for cotton and 21.338 mg·m$^{-2}$·s$^{-1}$ to 193.939 mg·m$^{-2}$·s$^{-1}$ for spring maize. Nevertheless, water vapor flux and $F_C$ showed opposite trends. The strongest changes in the water vapor flux occurred in July, followed by June and August, when spring maize had a higher peak value than cotton.

In 2018, the cotton yield in the research area was 2261 kg·hm$^{-2}$, with a total water of 5284 m$^3$·hm$^{-2}$, including precipitation of 1159 m$^3$·hm$^{-2}$ and irrigation of 4125 m$^3$·hm$^{-2}$. The WUE$_e$ of cotton was 0.43. The yield of spring maize is 12,891 kg·hm$^{-2}$, with a total water of 6607 m$^3$·hm$^{-2}$, including precipitation of 907 m$^3$·hm$^{-2}$ and irrigation of 5700 m$^3$·hm$^{-2}$. The WUE$_e$ of spring maize was 1.95. As crops do not undergo photosynthesis at night, our study of daily WUE$_l$ only considered daytime activity (10:00–20:00;

Figure 5). Overall, the $WUE_l$ of cotton was between $-6.16$ g·kg$^{-1}$ and $-4.35$ g·kg$^{-1}$. On a daily scale, $WUE_l$ intensity weakened gradually with increasing temperature and light. On a monthly scale, the peak occurred in July, whereas the weakest level occurred in June. For spring maize, the $WUE_l$ was between $-7.40$ g·kg$^{-1}$ and $-5.84$ g·kg$^{-1}$, slightly higher than that of cotton. We did not observe a day-to-day trend of the $WUE_l$ with respect to changes in temperature and light. On a monthly scale, the peak $WUE_l$ of spring maize occurred in July, whereas the weakest level was recorded in August.

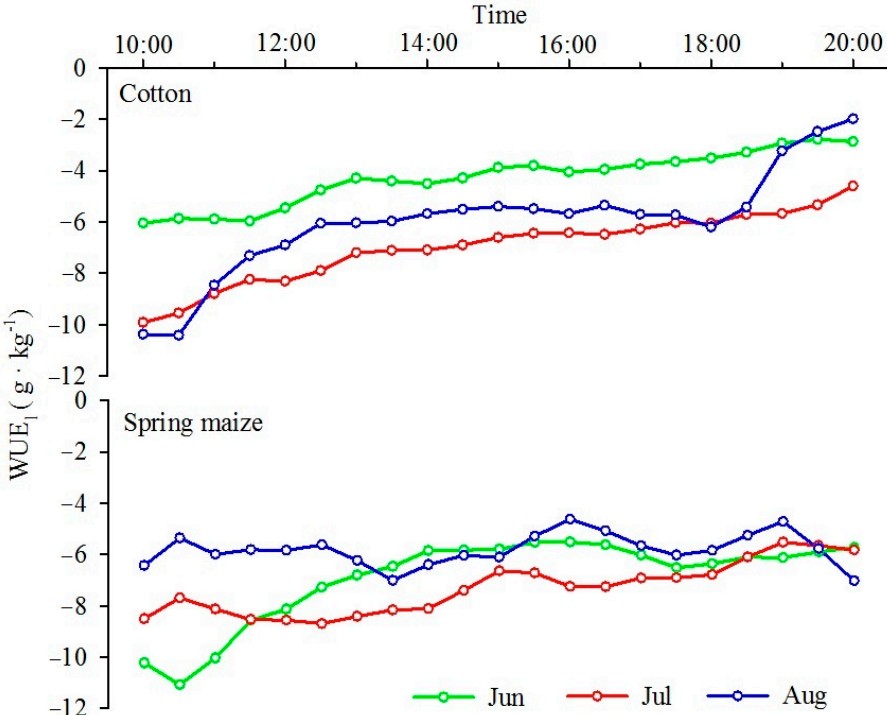

**Figure 5.** Monthly average daily variation in water use efficiency ($WUE_l$) in cotton and spring maize.

Our correlation analyses of $WUE_l$ and $T_a$, VPD, and Rn revealed that the main factors affecting the $WUE_l$ intensity were VPD and $T_a$, while Rn had relatively less influence on the $WUE_l$ intensity (Table 4). Further fitting of the nonlinear relationship among $WUE_l$ intensity, VPD, and $T_a$ in cotton and spring maize (Figure 6) showed that the $WUE_l$ intensity decreased gradually as the VPD increased because VPD reflects the water potential of leaves, and an increase in leaf water potential causes stomatal closure, leading to a reduction in $WUE_l$.

**Table 4.** Correlation coefficient (R) values for the relationships between $WUE_l$ and environmental factors in cotton and spring maize.

|  | $T_a$ | VPD | Rn |
|---|---|---|---|
| $WUE_l$ of Cotton | 0.418 * | 0.577 ** | $-0.039$ |
| $WUE_l$ of Spring maize | 0.456 ** | 0.449 * | 0.077 |

* $p < 0.05$; ** $p < 0.01$.

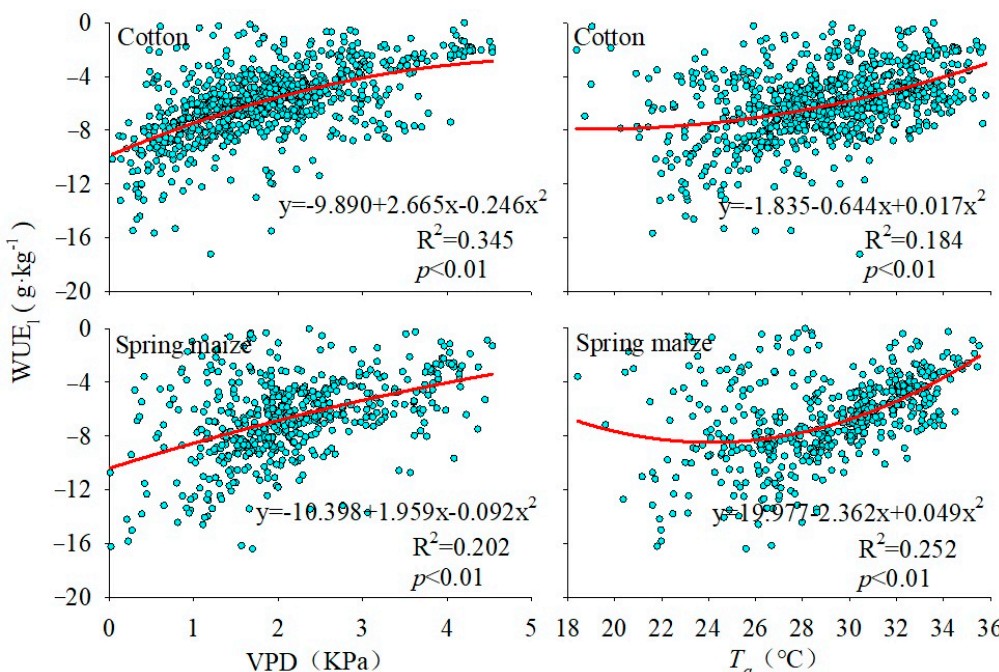

**Figure 6.** Relationship among $WUE_l$, saturated vapor pressure difference (VPD), and $T_a$ in cotton (**upper**) and spring maize (**lower**).

## 4. Discussion

In this study, we installed an eddy covariance system at the interface between adjacent cotton and spring maize fields and segmented the observed $CO_2$ and $H_2O$ turbulent flux based on the flux source partitioning analysis. We used this system to compare and analyze the differences in carbon and water exchange characteristics, WUE, and their relationship with environmental factors in cotton and spring maize fields. Our research effectively mitigated the systematic observation errors among different instruments and saved observation costs. Our findings also provide valuable references for subsequent studies on farmland and atmospheric observations.

Throughout the growing season, farmland H and LE fluxes exhibited high-low-high and low-high-low trends, respectively, displaying evident complementary effects. It was jointly influenced by solar radiative forcing and crop growth. As solar radiation varied in the growing season, the energy received by the surface showed a low-high-low trend, and crop growth gradually flourished, with the coverage of the ground reaching its peak in July. Meanwhile, the H flux reached its minimum value in the growing season, whereas the LE flux peaked due to the combined effects of plant transpiration and soil evaporation. Subsequently, as solar radiation weakened and crop growth declined, the LE flux gradually decreased, and the H flux rebounded to some extent.

Cotton and maize are C3 and C4 plants, respectively, with distinct $CO_2$ properties during photosynthesis. Unlike C3 plants, C4 plants have an extra $CO_2$ sequestration pathway, which provides a relatively higher $CO_2$ concentration for the C3 pathway in the vascular bundle sheath, and this process enhances the assimilation capacity of C4 plants compared to C3 plants [45]. As a result, spring maize has greater carbon sequestration capacity than cotton under the same conditions. Based on the flux data of this study, the NEE of cotton and spring maize throughout the entire growth period were $-321.2 \text{ g C·m}^{-2}$ and $-549.6 \text{ g C·m}^{-2}$, respectively. Compared with NEE data of other ecosystems at home and abroad (Table 5), the results of our research showed a high level of carbon balance, regardless of whether it was in a monoculture or a rotation farmland ecosystem, or a forest and grassland ecosystem. In contrast to forest and grassland ecosystems, farmland ecosystems are susceptible to the effects of agricultural activities and human disturbances, resulting in significant temporal and spatial variations in carbon and water fluxes [46]. Our

research indicates that temperature and net radiation are the main factors influencing the carbon sequestration capacity of cotton and spring maize. With the increase in temperature and net radiation, the carbon sequestration capacity of the two crops gradually increases. After reaching the optimal conditions, the increases in carbon sequestration efficiency slow down. The optimal temperature ranges for cotton and spring maize are 20 °C–25 °C and 22 °C–27 °C, respectively. When the temperature is lower than the optimal range, plants tend to be in their early stages or affected by cold stress, resulting in a lower carbon sequestration efficiency. On the contrary, when the temperature is higher than the optimal temperature, plants experience heat stress, leading to stomatal closure and a lower carbon sequestration efficiency.

Absorbing more $CO_2$ based on the consumption of unit water will greatly improve water use efficiency. Previous studies reported that there were differences in the carbon sequestration and water consumption capacities of ecosystems due to differences in crops, leading to significant differences in their WUE [4,25,32,47,48]. Therefore, spring maize, as a typical C4 plant, exhibits relatively higher water use efficiency compared to cotton at both leaf and ecosystem levels. Additionally, the highest carbon sequestration rate and water use efficiency occurred in July. VPD and $T_a$ were the two main factors influencing WUE among external environmental conditions. Plant water consumption increases as VPD increases, which gradually weakens WUE. The relationship between $T_a$ and WUE is a quadratic function, and too low and too high temperatures are both unfavorable for the absorption of $CO_2$ by crops, resulting in a decrease in their water use efficiency. This finding further underscores the significance of identifying the optimal temperature range for crops.

Farmland ecosystems can improve yields by changing planting systems, fertilization, and irrigation procedures, as well as other human-controlled measures. In terms of changes in carbon sequestration flux, spring maize reaches the maximum carbon sequestration state in July, the same time when spring maize reaches the harvesting conditions for ensiling feed. Therefore, it is recommended to immediately plant winter crops like winter wheat or winter vegetables after harvesting spring maize at the end of July to maximize the utilization of water and heat resources in farmland. And farmland rotation not only increases farmland production but also maximizes the carbon sequestration capacity of farmlands. To ensure crop yield sustainability, which is of great significance for the rational allocation and effective utilization of water resources and economic development in desert oases, it is necessary to understand the changing characteristics of $CO_2$ and water/heat transfer pathways, as well as consumption methods in oasis farmland ecosystems, and explore effective management models to improve WUE.

**Table 5.** Comparison of net ecosystem carbon exchange (NEE) in different ecosystems.

| Site | Ecosystem Type | NEE/ (g C·m$^{-2}$) | Research Period | Source |
|---|---|---|---|---|
| Ulan Wusu, China | Farmland (cotton) | −478.6 | April–October 2009–2010 | [49] |
| Washington, USA | Farmland (wheat) | −261 | 2012–2013 | [50] |
| Nebraska, USA | Farmland (maize) | −590.0 | May–October 2002 | [51] |
| Yingke, China | Farmland (maize) | −626.0 | 2008–2009 | [52] |
| Weishan, China | Farmland (wheat, maize) | −533 ~ −585 | 2006–2008 | [24] |
| Central Oregon, USA | Forest | −534 ~ −415 | 2004–2008 | [40] |
| Northeast China | Grassland | −89.57 | 2014 | [47] |

## 5. Conclusions

In this study, we utilized a set of eddy covariance installed at the interface between cotton and spring maize crops in arid regions to analyze the differences in carbon and water fluxes between the two typical crops, as well as their responses to changes in environ-

mental factors. Our research effectively mitigated the systematic observation errors among different instruments and saved observation costs. The research results show that spring maize, as a typical representative of C4 plants, has a higher carbon sequestration capacity and water use efficiency than cotton. The NEE of cotton and spring maize throughout their entire growth period were $-321.2$ g C m$^{-2}$ and $-549.6$ g C m$^{-2}$, respectively. Therefore, the carbon sequestration capacity of farmland ecosystems in arid regions is undeniable and is greatly influenced by the growth and development of plants. Throughout the growth period, temperature, net surface radiation, and VPD significantly affected the carbon sequestration capacity and WUE of cotton and spring maize. In addition, based on the changes in the carbon flux of spring maize, it was recommended to immediately plant winter crops like winter wheat or winter vegetables after harvesting spring maize at the end of July to maximize the utilization of water and heat resources in farmlands. Moreover, farmland rotation not only increases farmland production but also maximizes the carbon sequestration capacity of farmlands.

**Author Contributions:** Conceptualization, X.Z. and F.Y.; methodology, F.Y. and X.Z.; software, J.G.; validation, X.Z., F.Y. and A.M.; formal analysis, X.Z. and C.J.; investigation, X.Y. and Y.W.; resources, F.Y.; data curation, F.Y., Y.S., J.G. and M.M.; writing—original draft preparation, X.Z.; writing—review and editing, X.Z., X.H., W.W., Z.C. and C.L.; visualization, W.H., C.Z. and S.A.; supervision, C.J.; project administration, F.Y. and A.M.; funding acquisition, F.Y. All authors have read and agreed to the published version of the manuscript.

**Funding:** This work was jointly supported by the Natural Science Foundation of Xinjiang Uygur Autonomous Region (2022D01E104), the Chinese Desert Meteorological Science Research Found (Sqj 2018007), the National Natural Science Foundation of China (41975010), and the Scientific and Technological Innovation Team (Tianshan Innovation Team) Project of Xinjiang (Grant No. 2022TSYCTD0007).

**Data Availability Statement:** The data used in this paper can be provided by F.Y. (yangfan309@yeah.net) upon request.

**Conflicts of Interest:** The authors declare no conflict of interest.

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
