# Peer review of "Farmland Carbon and Water Exchange and Its Response to Environmental Factors in Arid Northwest China"

_land, doi:10.3390/land12111988_

Round 1

Reviewer 1 Report

In the global carbon cycle, the terrestrial ecosystem is an important carbon sink and source. Farmland ecosystem is one most active part. So, farmland ecosystem is an important for the carbon cycle. C3 and C4 different species crops have different using carbon policies in the different geographical areas. In the arid-semi arid areas, maybe, it is more obvious. In this MS, authors compared cotton and spring maize in the Xinjiang based on the eddy covariance method to make clear carbon factors of farmland ecosystems and its relationship with environmental and human managements. It is useful to understand deeply the farmland ecosystem carbon cycle.

Questions and Suggestions:

1.       In the “Site description”, it is better to introduce more about soil characteristics, evaporation, precipitation, etc., with the relationship of farming and vegetation.

2.       From the figure 1, weather observation system sets up in the middle of cotton field on the west and maize field on the east, these two fields are next to each other. So, whether or not have the data about the cotton and maize fields representativeness and accurate?

3.       It is better to introduce simply the eddy covariance system, including type, manufacturer, etc.

4.       Lines 175-176, “??and ??are the fluctuating values of water vapour concentration and CO2 concentration, respectively,”, the second ??is ?C.

5.       ?cis the fluctuating value CO2 concentration, that means, it is CO2 concentration in the air, and line 171, in the “CO2 (FC)”, Fc stands for the CO2 concentration, what kind difference between these tow ?cand FC? From the following, Fc is CO2 flux value, So, it is best to describe at the first appearing site.

6.       CO2 concentration comes from the device determining, based on some sensors, whether or not authors have tested based on some GC device or other devices?  

7.       In the figure 2, “Corn” in the figure is best to change to “Spring Maize”, keeping the same with the text.

8.       Line 208, “……, regardless of the crop.”, what is means about this? all different species crops?

9.       Lines 320-325, “Cotton and maize are C3 and C4 plants, respectively, with distinct CO2 properties during photosynthesis [36]. Unlike C3 plants, C4 plants have an extra CO2 fixation pathway, which provides a higher CO2 concentration for the C3 pathway in the vascular bundle sheath; this process enhances the assimilation capacity of C4 plants compared to C3 plants [37]. As a result, under comparable conditions, maize has a greater carbon fixation ability than cotton, and exhibits relatively consistent daily WUE.”, C3 and C4 have a carbon fixation difference from the results in this MS, but, there are a opposite tendency about monthly average daily variation in CO2 flux (FC) for cotton and spring maize in the figure 3, especially, in the morning of July and August. Why does it like this?

Moderate editing of English language required.

Author Response

Response to Reviewer #1's Comments:

 In the global carbon cycle, the terrestrial ecosystem is an important carbon sink and source. Farmland ecosystem is one most active part. So, farmland ecosystem is an important for the carbon cycle. C3 and C4 different species crops have different using carbon policies in the different geographical areas. In the arid-semi arid areas, maybe, it is more obvious. In this MS, authors compared cotton and spring maize in the Xinjiang based on the eddy covariance method to make clear carbon factors of farmland ecosystems and its relationship with environmental and human managements. It is useful to understand deeply the farmland ecosystem carbon cycle.

Response: We really appreciate Reviewer #1 for giving full recognition to the manuscript.

 Specific comments

  1. In the “Site description”, it is better to introduce more about soil characteristics, evaporation, precipitation, etc., with the relationship of farming and vegetation.

Response: According to this comment, we added relevant content such as soil characteristics, evaporation, precipitation, etc. to the “Site description” in the lines 131-136 of the revised manuscript.

  1. From the figure 1, weather observation system sets up in the middle of cotton field on the west and maize field on the east, these two fields are next to each other. So, whether or not have the data about the cotton and maize fields representativeness and accurate?

Response: According to the experimental design, we installed a set of eddy covariance system at the interface of cotton and spring maize fields, and split the observed CO2 turbulent fluxes based on the flux source area. The westerly turbulent flux represented cotton, and the easterly turbulent flux represented spring maize. This approach ensures the representativeness and accuracy of the data.

  1. It is better to introduce simply the eddy covariance system, including type, manufacturer, etc.

Response: According to this comment, we have added Table 2 in the revised manuscript to introduce the basic information of the experimental observation systems, including the model, manufacturer, observation variables, and installation location of each sensor.

  1. 4. Lines 175-176, “ρν′ and ρν′ are the fluctuating values of water vapour concentration and CO2 concentration, respectively,”, the second ρν′ is ρc′.

Response: According to this comment, we have modified in the line 197 of the revised manuscript.

  1. ρc′ is the fluctuating value CO2 concentration, that means, it is CO2 concentration in the air, and line 171, in the “CO2 (FC)”, Fc stands for the CO2 concentration, what kind difference between these tow ρc′ and FC? From the following, FC is CO2 flux value, So, it is best to describe at the first appearing site.

Response: According to this comment, the CO2 flux (FC) first appeared in line 166 of the revised manuscript, we have been annotated in the revised manuscript.

  1. CO2 concentration comes from the device determining, based on some sensors, whether or not authors have tested based on some GC device or other devices?

Response: In this study, we used an infrared gas analyzer of the eddy covariance system to measure CO2 concentration. To ensure the accuracy of the observation data, we conducted a complete calibration test on the eddy covariance equipment before its installation. Firstly, we calibrated the observation zero point of the infrared gas analyzer for CO2 and H2O observation using a zero point generator. Secondly, we calibrated the CO2 span of the analyzer using standard carbon dioxide gas. Finally, we calibrated the H2O observation span of the analyzer using a dew point generator. After completing all of the above calibrations, the eddy covariance system can effectively reflect the CO2/H2O exchange in farmland.

  1. In the figure 2, “Corn” in the figure is best to change to “Spring Maize”, keeping the same with the text.

Response: According to this comment, we have changed 'corn' to 'spring maize' in the revised manuscript.

  1. Line 208, “……, regardless of the crop.”, what is means about this? all different species crops?

Response: According to this comment, “the crop” refers to spring maize and cotton. We have modified as “……, there was a significant complementary effect between H and LE of spring maize and cotton during the growth period.” in the lines 230-231 of the revised manuscript.

  1. Lines 320-325, “Cotton and maize are C3 and C4 plants, respectively, with distinct CO2 properties during photosynthesis [36]. Unlike C3 plants, C4 plants have an extra CO2 fixation pathway, which provides a higher CO2 concentration for the C3 pathway in the vascular bundle sheath; this process enhances the assimilation capacity of C4 plants compared to C3 plants [37]. As a result, under comparable conditions, maize has a greater carbon fixation ability than cotton, and exhibits relatively consistent daily WUE.”, C3 and C4 have a carbon fixation difference from the results in this MS, but, there are a opposite tendency about monthly average daily variation in CO2 flux (FC) for cotton and spring maize in the figure 3, especially, in the morning of July and August. Why does it like this?

Response: According to this comment, we have conducted quality control and screening of the data again. The screened data has greatly improved, especially by removing the unexplained negative CO2 flux of spring maize at night in July and August in Figure 3.

Reviewer 2 Report

This article is well written and has practical value. The figures and tables are beautifully done and have a high academic level.

In order to help the authors, here are some comments that, hopefully, will be beneficial to the authors to improve their manuscript:

1. Please introduce the basic situations of the plots, including growth period, plant height, density, growth potential, etc.

2. How environmental factors are collected, including the location, method and frequency of collection, etc.

3. How is carbon and water exchange reflected in the experimental design?

4. In Figure 4, it is suggested to add P-values.

5. In the results, the response of carbon and water exchange to environmental factors can be further strengthened, and the expression needs to be more clear.

6. The conclusions should be simplified, and the content of the result description should be presented in the results.

Author Response

Response to Reviewer #2's Comments:

This article is well written and has practical value. The figures and tables are beautifully done and have a high academic level.

Response: We really appreciate Reviewer #2 for giving full recognition to the manuscript.

Specific comments

  1. Please introduce the basic situations of the plots, including growth period, plant height, density, growth potential, etc.

Response: According to this comment, we have added Table 1 to introduce the basic situations of the plots, including plant growth period, plant height, and irrigation amount.

  1. How environmental factors are collected, including the location, method and frequency of collection, etc.

Response: According to this comment, we have added Table 2 in the manuscript to introduce the collection methods and information of environmental factors, including various environmental factors, sensors models and manufacturers, installation locations, and data collection frequency.

  1. How is carbon and water exchange reflected in the experimental design?

Response: According to the experimental design, we installed a set of eddy covariance system at the interface of cotton and spring maize fields, and split the observed CO2/H2O turbulent fluxes and based on the flux source area. The westerly turbulent flux represented cotton, and the easterly turbulent flux represented spring maize. Then, compare and analyze the differences in carbon and water exchange between cotton and spring maize.

  1. In Figure 4, it is suggested to add P-values.

Response: According to this comment, we have added P-values in both Figure 4 and Figure 7, respectively.

  1. In the results, the response of carbon and water exchange to environmental factors can be further strengthened, and the expression needs to be more clear.

Response: According to this comment, we have optimized the description of how environmental factors affect farmland carbon and water flux in the results section. In addition, we have elaborated on the details of the above results during the discussion in the revised manuscript.

  1. The conclusions should be simplified, and the content of the result description should be presented in the results.

Response: According to this comment, we have rewritten the conclusions section of the revised manuscript.

Reviewer 3 Report

1.The abstract of the paper suggests that "Our results indicated that the yield and carbon sequestration capacity of farmland ecosystems can be improved using appropriate field management measures, thereby playing an important supporting role in global carbon storage." However, in the discussion and conclusion, it is not clear which specific field management measures can effectively enhance the carbon sequestration ability of cotton, spring maize or similar agroecosystems, so it is suggested to supplement and improve the paper to make the significance of the research clearer.

2.This manuscript proposes that net surface radiation, temperature, saturated vapor pressure and other factors are the main factors affecting the carbon sequestration ability of crops. However, the field management measures has a limited impact on these natural factors, which are more likely to affect the pattern of agricultural distribution, planting structure, and it is suggested to combine the conditions of the study area to put forward the measures and suggestions to enhance the carbon sequestration ability of the agricultural system.

Moderate editing of English is needed for publishing.

Author Response

Response to Reviewer #3's Comments:

  1. The abstract of the paper suggests that "Our results indicated that the yield and carbon sequestration capacity of farmland ecosystems can be improved using appropriate field management measures, thereby playing an important supporting role in global carbon storage." However, in the discussion and conclusion, it is not clear which specific field management measures can effectively enhance the carbon sequestration ability of cotton, spring maize or similar agroecosystems, so it is suggested to supplement and improve the paper to make the significance of the research clearer.

Response: According to this comment, we have modified the discussion and conclusions of the revised manuscript. The research results show that corn in July showed the highest carbon sequestration capacity and also met the harvesting requirements for silage feed. We proposed that spring maize could be harvested at the end of July as silage feed. Afterward, winter crops such as winter wheat or winter vegetables could be planted to maximize the utilization of water and heat resources in the farmland. Rotation could improve the carbon sequestration capacity of farmland ecosystems.

  1. This manuscript proposes that net surface radiation, temperature, saturated vapor pressure and other factors are the main factors affecting the carbon sequestration ability of crops. However, the field management measures has a limited impact on these natural factors, which are more likely to affect the pattern of agricultural distribution, planting structure, and it is suggested to combine the conditions of the study area to put forward the measures and suggestions to enhance the carbon sequestration ability of the agricultural system.

Response: In previous studies, the cropping regionalization of cotton and spring maize focused more on thermal conditions due to its irrigation-based agriculture in Xinjiang. However, the results of this study could provide new references for the determination of regionalization factors. In the future, it would be beneficial to conduct observations at multiple locations, which would lead to more accurate results for the next round of agricultural climate resource zoning.

Reviewer 4 Report

Land-2577337

Title: Farmland carbon and water exchange and its response to environmental factors in arid Northwest China

Authors: X. Zheng, F. Yang, A. Mamtimin, X. Huo, J. Gao, C. Ji, ..., & X. Yang

Date: 28 August, 2023

This paper shows carbon, energy, and water fluxes for cotton and maize crops in an arid, but irrigated, region.  The study uses eddy covariance observations for a single growing season, where a single measurement station was placed between cotton and maize fields.  The authors show the resulting carbon fluxes, as well as fluxes of latent and sensible heat; and they also correlate these fluxes to environmental factors (temperature, vapor pressure deficit, and radiation).  While it is interesting to see a direct comparison of maize (C4) and cotton (C3) land-atmosphere exchanges, due to methodological issues and repetitive content, I cannot recommend this paper for publication as it stands now.

Premise and Introduction

·      This study aims to show the carbon sequestration ability of cotton and maize; however, the productivity of these crops has already been well-established. 

·      This study does not provide adequate review of previous work, thus the results shown here are duplicative to previous efforts.  For example, they state that there is a lack of detailed farmland analysis, yet there exists a large volume of work on carbon and energy cycles for croplands.

·      Here is a quick list of just some of the publications that have specifically looked at maize productivity and water (and energy) fluxes:

o   Bassu et al. (2014).  How do various maize crop models vary in their responses to climate change factors? Global Change Biol.

o   Cheng et al. (2013).  Simulating greenhouse gas mitigation potential for Chinese croplands using the DAYCENT ecosystem model.  Global Change Biol.

o   Corbin et al., 2010: Assessing the impact of crops on regional CO2 fluxes and atmospheric concentrations.  Tellus B.

o   Fang et al., 2013.  Modeling evapotranspiration and energy balance in a wheat-maize cropping system using the revised RZ-SHAW model.  Ag. For. Meteorol.

o   Li et al., 2007.  Latent and sensible heat fluxes and energy balance in a maize agroecosystem.  J. Plant Ecology.

o   Li et al., 2010.  Evaluating the crop water stress index and its correlation with latent heat and CO2 fluxes over winter wheat and maize in the North China Plain.  Agric. Water Manage.

o   Lokupitiya et al., 2009.  Incorporation of crop phenology in Simple Biosphere Model to improve land-atmosphere exchanges from croplands.  Biogeosciences.

o   Verma et al., 2005.  Annual carbon dioxide exchange in irrigate and rainfed maize-based agroecosystems.  Agric. Forest Meteorol.

o   Vleeshouwers and Verhagen, 2002.  Carbon emission and sequestration by agricultural land use: a model study for Europe.  Global Change BIol.

o   Zhang et al., 2002.  Variation of fluxes of water vapor, sensible heat, and carbon dioxide above winter wheat and maize canopies.  J. Geographical Sciences.

o   Zhang, 2004.  Effect of soil water deficit on evapotranspiration, crop yield, and water use efficiency in the North China Plain.  Agric. Water Manage.

·      Here is a quick list of some of the publications on cotton:

o   Bezerra et al. (2015).  Surface energy exchange and evapotranspiration from cotton crop under full irrigation conditions in the RIo Grande do Norte State, Brazilian Semi-Arid.  Bragantia.

o   Haymann et al. (2019).  Effects of variable fetch and footprint on surface renewal measurements of sensible and latent heat fluxes in cotton.  Ag. For. Meteorol.

o   Li et al. (2018) Relationships between carbon fluxes and environmental factors in a drip-irrigated, film-mulched cotton field in arid region.  PLoS One.

o   Liu and Qiao, 2023.  Partitioning evapotranspiration in a cotton field under mulched drip irrigation based on water-carbon fluxes coupling in an arid region in Northwestern China.  Agriculture.

o   Rosa and Tanny, 2015.  Surface renewal and eddy covariance measurements of sensible and latent heat fluxes of cotton during two growing seasons. Biosyst. Eng.

o   Tolk et al. (2006).  Nighttime evapotranspiration from alfalfa and cotton in a semi-arid climate.  Agronomy.

Methods

·      The authors negated to state the amount and methodology of irrigation.  This information needs to be provided, as it vasty impacts the crop fluxes.  See Verma et al. (2005) as an example.

·      The authors show a picture of the study area and a wind direction frequency plot; however, to my understanding, they did not use this information to separate the fluxes between the two different crops.  Since the instruments are only 3 m above the ground, they are influenced by the local fluxes, and the amount of influence from the maize and cotton would change depending on wind speed.  They did not address this, nor explain how they separated out the fluxes from the two fields with a single measurement tower, which needs to be done to trust that the results between the two crops are independent. 

Results

·      The authors showed monthly mean diurnal variations of sensible and latent heat fluxes; however, the figures and text showed well-established results and described them in a way that did not incorporate the already-known information or provide new information from previous work.  It is already known that crops have a strong seasonal cycle in latent and sensible heat fluxes, with the energy fluxes trading off and switching to latent heat flux during the peak growing season.  The authors described this phenomenon as being U or single-peak shaped, rather than relating this back to the biophysical phenomenon.

·      The authors often use the term "in different stages", yet they do not describe these, nor do they separate out the fluxes except by month.  Crops have well-defined growing stages, and to make their statement, the results should be parsed by growing stage rather than (or in addition to) by month.

·      The authors show diurnal variations of CO2 flux and its relationship to meteorological factors; however, again this work has already been done extensively and it is well known that radiation and temperature influence crop growth.  It is no surprise that VPD does not have a strong relationship, since it finally came out in this section that these crops are irrigated:  crops without water stress are not going to be impacted by water vapor deficit, since they do not have a water deficit.

·      The authors show monthly average daily variation of water vapor flux, but this is the same as latent heat flux, making this section duplicate.  On top of this, the figure shown is labeled as sensible heat flux. 

Discussion

·      The authors discuss how different farming procedures can have an impact on potential carbon sequestration, yet they did not offer any support as to how in this manuscript, thus they did not support this statement.  They claim that the use of drip technology conserves water; however, this needs to be shown by comparing fluxes and yield between crops using drip and a control plot (without or with different irrigation).  They did offer one comparison to previously published results, but the research period was a different year, and the site was at a different location, thus the cause of the NEE changes cannot be definitively concluded.

Conclusions

·      The authors give exact amounts of carbon and water fluxes and timing; however, since they did not state how they separated out the fluxes between these two different crops with a single measurement, these conclusions are not trustworthy or robust. 

Author Response

Response to Reviewer #4's Comments:

This paper shows carbon, energy, and water fluxes for cotton and maize crops in an arid, but irrigated, region.  The study uses eddy covariance observations for a single growing season, where a single measurement station was placed between cotton and maize fields.  The authors show the resulting carbon fluxes, as well as fluxes of latent and sensible heat; and they also correlate these fluxes to environmental factors (temperature, vapor pressure deficit, and radiation).  While it is interesting to see a direct comparison of maize (C4) and cotton (C3) land-atmosphere exchanges, due to methodological issues and repetitive content, I cannot recommend this paper for publication as it stands now.

Response: Thanks for reviewing the manuscript. We accept all of the suggestions provided. The revised manuscript has added many details of the experimental design. Additionally, it provides practical farming management recommendations based on the observed results. The quality of the revised manuscript has greatly improved.

Specific comments

  1. Premise and Introduction

This study aims to show the carbon sequestration ability of cotton and maize; however, the productivity of these crops has already been well-established.

This study does not provide adequate review of previous work, thus the results shown here are duplicative to previous efforts. For example, they state that there is a lack of detailed farmland analysis, yet there exists a large volume of work on carbon and energy cycles for croplands.

Response: According to this comment, we have further summarized and organized the previous research on carbon and water flux in farmland. In the introduction of the manuscript, we have supplemented research progress and literature information, and succinctly condensed the scientific questions and significance.

  1. Methods

The authors negated to state the amount and methodology of irrigation.  This information needs to be provided, as it vasty impacts the crop fluxes.  See Verma et al. (2005) as an example.

The authors show a picture of the study area and a wind direction frequency plot; however, to my understanding, they did not use this information to separate the fluxes between the two different crops. Since the instruments are only 3 m above the ground, they are influenced by the local fluxes, and the amount of influence from the maize and cotton would change depending on wind speed.  They did not address this, nor explain how they separated out the fluxes from the two fields with a single measurement tower, which needs to be done to trust that the results between the two crops are independent.

Response: According to this comment, we have supplemented the information on irrigation in the manuscript. As shown in Figure 1, we installed an eddy covariance system at the interface between cotton and spring maize fields. Spring maize is relatively tall, reaching up to approximately 2 meters during their growth stage. Under the condition that the installation height of the instrument is more than 1.5 times the plant height, we installed the eddy covariance system at a height of 3 meters above the ground. The corresponding flux range is approximately within the range of 300 m. The flux contribution range will be influenced by wind speed. However, in this study, the half hour average wind direction observed at the interface is the only indicator that distinguishes flux information between different crops. Different crops were distinguished based on different wind directions. After data quality control, a total of 7731 effective flux data were collected during the entire observation period. Among them, 4,099 information was from the east, representing the spring maize, and 3,632 information was from the west, representing the cotton. This approach ensures the representativeness and accuracy of the data.

  1. Results

The authors showed monthly mean diurnal variations of sensible and latent heat fluxes; however, the figures and text showed well-established results and described them in a way that did not incorporate the already-known information or provide new information from previous work. It is already known that crops have a strong seasonal cycle in latent and sensible heat fluxes, with the energy fluxes trading off and switching to latent heat flux during the peak growing season.  The authors described this phenomenon as being U or single-peak shaped, rather than relating this back to the biophysical phenomenon.

Response: There have been many studies on heat flux in farmland, and our results are similar to those of previous research findings. However, as one of the essential component in flux studies, the analysis of heat flux cannot be overlooked. This also provides valuable background information on the crop growth stage for subsequent research.

The authors often use the term "in different stages", yet they do not describe these, nor do they separate out the fluxes except by month. Crops have well-defined growing stages, and to make their statement, the results should be parsed by growing stage rather than (or in addition to) by month.

Response: The manuscript compared and analyzed the carbon and water fluxes of crops with two different growth stages. Since the two different stages cannot be completely matched, we conducted a comparative analysis of the flux differences between the two crops on a monthly scale in the manuscript.

The authors show diurnal variations of CO2 flux and its relationship to meteorological factors; however, again this work has already been done extensively and it is well known that radiation and temperature influence crop growth. It is no surprise that VPD does not have a strong relationship, since it finally came out in this section that these crops are irrigated:  crops without water stress are not going to be impacted by water vapor deficit, since they do not have a water deficit.

Response: We have adopted the comment, and added relevant content in the lines 260-261 of the manuscript.

The authors show monthly average daily variation of water vapor flux, but this is the same as latent heat flux, making this section duplicate. On top of this, the figure shown is labeled as sensible heat flux.

Response: According to this comment, we have reduced the description of water vapor flux and deleted Figure 5 in the revised manuscript.

  1. Discussion

The authors discuss how different farming procedures can have an impact on potential carbon sequestration, yet they did not offer any support as to how in this manuscript, thus they did not support this statement. They claim that the use of drip technology conserves water; however, this needs to be shown by comparing fluxes and yield between crops using drip and a control plot (without or with different irrigation).  They did offer one comparison to previously published results, but the research period was a different year, and the site was at a different location, thus the cause of the NEE changes cannot be definitively concluded.

Response: Based on the analysis results, we have supplemented suggestions for farmland management and protection in the lines 372-379 of the revised manuscript. In addition, the manuscript provides a comparison of the carbon sequestration capacity of farmland, forest, and grassland ecosystems in different regions. The purpose is to provide readers with a overall conceptual system to position the carbon sequestration capacity of farmland in different ecosystems.

  1. Conclusions

The authors give exact amounts of carbon and water fluxes and timing; however, since they did not state how they separated out the fluxes between these two different crops with a single measurement, these conclusions are not trustworthy or robust.

Response: In the method section of the revised manuscript, we have added many details to strengthen support for the results of this paper. The conclusion of the revised manuscript is reliable.

Reviewer 5 Report

1. In one paragraph the novelty research should be explained. 

2. In figure 7 page 10 why R2 values is low please discuss about it.

Author Response

Response to Reviewer #5's Comments:

  1. In one paragraph the novelty research should be explained.

Response: According to this comment, we have reorganized the novelty of the paper and made modifications in lines 109-124 of the revised manuscript.

  1. In figure 7 page 10 why R2 values is low please discuss about it.

Response: Water use efficiency is the ratio of the net ecosystem carbon exchange to the evapotranspiration. Through correlation analysis, we have found that VPD and air temperature were the main factors affecting water use efficiency. However, water use efficiency is influenced by various environmental factors, such as temperature, photosynthetic active radiation, precipitation, soil moisture, etc., which may be the reason for the lower R2 value in regression analysis.

Reviewer 6 Report

In this paper, the carbon water exchange characteristics of two typical crops, cotton and spring maize, in the arid area of northern Xinjiang of China were assessed by the eddy covariance method, and the water carbon changes of crops and their responses to environmental factors during the growth cycle were evaluated. This is of great significance for water and carbon exchange and carbon neutral management of farmland ecosystems in arid areas. However, there are some major flaws in this paper that hinder its publication:

First of all, the introduction only emphasizes the importance of carbon sequestration in farmland ecosystems, and does not point out the shortcomings of current research and the reasons for conducting this study. None of this explains why you do the work, and the innovation of the work is not reflected. It would be wise to revisit the introduction and ask a good scientific question.

Second, the discussion part did not dig deeply into the results of data analysis, but only compared the fluxes of sensible heat (H) and latent heat (LE) and CO2 (FC) of cotton and spring maize. What is the difference between this part of discussion and the previous research results? What gaps are filled? The article is not explained, which makes the article incomplete.

In this paper, the carbon water exchange characteristics of two typical crops, cotton and spring maize, in the arid area of northern Xinjiang of China were assessed by the eddy covariance method, and the water carbon changes of crops and their responses to environmental factors during the growth cycle were evaluated. This is of great significance for water and carbon exchange and carbon neutral management of farmland ecosystems in arid areas. However, there are some major flaws in this paper that hinder its publication:

First of all, the introduction only emphasizes the importance of carbon sequestration in farmland ecosystems, and does not point out the shortcomings of current research and the reasons for conducting this study. None of this explains why you do the work, and the innovation of the work is not reflected. It would be wise to revisit the introduction and ask a good scientific question.

Second, the discussion part did not dig deeply into the results of data analysis, but only compared the fluxes of sensible heat (H) and latent heat (LE) and CO2 (FC) of cotton and spring maize. What is the difference between this part of discussion and the previous research results? What gaps are filled? The article is not explained, which makes the article incomplete.

 In addition, some specific comments are as follows:

1. This study mainly studied the water-carbon flux characteristics of two typical crops and their impact on environmental change, and the conclusion does not support the impact of different field management measures on the carbon sequestration capacity of farmland ecosystems. Therefore, the research significance of this work needs to be re-examined.

2. The introduction of this paper needs to be reorganized, the scientific problems and innovations proposed are not clear, and the logic is confused. Please supplement relevant previous research progress.

3.Line 175: there are two ?v’, please modify them.

4.Line 242-251: Figure 4 How to determine the optimal growth temperature of cotton and spring maize?

5. The discussion needs to rearrange the problems and arguments. The literature in many places cannot support your views, and many documents are irrelevant to the main content, such as [40];

6. Water use efficiency and CO2 flux and their influencing factors were not discussed in the discussion.

Author Response

Response to Reviewer #6's Comments:

In this paper, the carbon water exchange characteristics of two typical crops, cotton and spring maize, in the arid area of northern Xinjiang of China were assessed by the eddy covariance method, and the water carbon changes of crops and their responses to environmental factors during the growth cycle were evaluated. This is of great significance for water and carbon exchange and carbon neutral management of farmland ecosystems in arid areas. However, there are some major flaws in this paper that hinder its publication:

First of all, the introduction only emphasizes the importance of carbon sequestration in farmland ecosystems, and does not point out the shortcomings of current research and the reasons for conducting this study. None of this explains why you do the work, and the innovation of the work is not reflected. It would be wise to revisit the introduction and ask a good scientific question.

Second, the discussion part did not dig deeply into the results of data analysis, but only compared the fluxes of sensible heat (H) and latent heat (LE) and CO2 (FC) of cotton and spring maize. What is the difference between this part of discussion and the previous research results? What gaps are filled? The article is not explained, which makes the article incomplete.

Response: Firstly, we have further summarized and organized the previous research on carbon and water flux in farmland. In the introduction of the manuscript, we have supplemented research progress and literature information, and succinctly condensed the scientific questions and significance. Secondly, we added many details in the discussion section, especially we provided management suggestions for rotation in spring maize fields based on the analysis results. This can maximize the carbon sequestration capacity of farmland.

Specific comments

  1. This study mainly studied the water-carbon flux characteristics of two typical crops and their impact on environmental change, and the conclusion does not support the impact of different field management measures on the carbon sequestration capacity of farmland ecosystems. Therefore, the research significance of this work needs to be re-examined.

Response: According to this comment, we have modified the discussion and conclusions in the lines 372-379 and 410-415 of the revised manuscript, clearly proposing field management measures to enhance the carbon sequestration capacity of farmland ecosystems.

  1. The introduction of this paper needs to be reorganized, the scientific problems and innovations proposed are not clear, and the logic is confused. Please supplement relevant previous research progress.

Response: According to this comment, we have reorganized the introduction, and proposed the scientific problems and innovations of the revised manuscript.

  1. Line 175: there are two ρν′, please modify them.

Response: The second ρν′ should be ρc′, we have modified in the lines 196 of the revised manuscript.

  1. Line 242-251: Figure 4 How to determine the optimal growth temperature of cotton and spring maize?

Response: Based on the research results of this paper and the experience of agricultural technicians, the optimal growth temperature range of cotton was determined to be 20-25 ℃, and the optimal growth temperature range of spring maize was determined to be 22-27 ℃. We have modified in lines 266-274 of the revised manuscript.

  1. The discussion needs to rearrange the problems and arguments. The literature in many places cannot support your views, and many documents are irrelevant to the main content, such as [40];

Response: According to this comment, we have reorganized the problems and arguments. The revised manuscript has been supplemented with literature information and condensed on the scientific issues studied.

  1. Water use efficiency and CO2 flux and their influencing factors were not discussed in the discussion.

Response: According to this comment, we have added the discussions on CO2 flux and water use efficiency and their influencing factors in the lines 347-357 and 364-370 of the revised manuscript.

Round 2

Reviewer 4 Report

This paper shows the carbon and water exchanges of maize and cotton in an arid environment in China, as well as the leaf-level water-use response to atmospheric variables.  While the authors provided written responses to the review comments, I do not believe they fully addressed the problems.  The major problems outlined previously remain: 1) the introduction does not adequately review previous work, 2) the uniqueness of this work is not clearly stated, and 3) they did not address the total use of water and its impact on canopy to ecosystem scale WUE and water resources, as they only addressed leaf-scale WUE.

1. Premise and Introduction

While the authors altered the introduction, much of this was rewording rather than getting at the main problem with the introduction, which was that there is not an adequate review of previous literature related to this work.  Reading through the introduction, there are many references to articles that are not appropriate or that need to be updated, and this is on top of not including an adequate discussion of previously-published work that is remarkably similar.

·      Lines 49-51: "Farmland ecosystems account for ~11% of global land area and are the most active part of the global ecosystem carbon pool [6-7]."  First off, [6] is a 1997 study about agricultural soil carbon and [7] is about CO2 exchanges over China (although the DOI was incorrect and I was not able to obtain this reference).  Neither of these papers are recent studies on global agricultural land area.  Second, this information is obtained annually by the FAO (Food and Agricultural Organization of the United Nations), including global trends of agricultural land area (https://www.fao.org/sustainability/news/detail/en/c/1274219/).  According to the latest reports, cropland is decreasing per capita, with 10% being used for permanent crops (including fruit trees, oil palm plantations, and cocoa plantations).  Third, [6] says "By itself, C sequestration in agricultural soils can make only modest contributions (e.g. 3-6% of total fossil C emissions) to mitigating greenhouse gas emissions."  This statement is in direct disagreement with what is written in the manuscript.

·      Lines 52-53: "Studies have shown that 20% of CO2 in the atmosphere comes from agricultural activity and related production processes [8]."  [8] is a 1999 paper talking about the potential of world cropland soils to sequester C.  With the rapid changes in carbon emissions, the statement in the manuscript needs to be supported by a more recent and relevant publication. 

·      Lines 54-56: "The carbon cycle in farmland ecosystems is very complex..." The reference cited, [9], is a modelling paper from 2015, which has this very phrase.  With all the work on the carbon cycle and crops, we cannot say this anymore as there has been vast literature on this.  For example, there are now even one-to-two page summaries on the crop carbon cycle, see http://nmsp.cals.cornell.edu/publications/factsheets/factsheet91.pdf for example. 

·      Lines 60-62:  [8] states the "total potential of C sequestration from croplands is 0.75-1 Pg/yr or ~50% of annual emissions of 1.6-1.8 Pg by deforestation and other agricultural activities," which is almost the text in the manuscript verbatim.  Again, with the recent changes in 1) agricultural land cover and 2) crop productivity, more recent statistics are available and should be referenced.

·      Lines 80-97: The manuscript discusses how "good" the eddy covariance technique is, yet never mentions any of the flaws in the measuring technique, such as energy balance closure, which are stated in the references they include [14] and [15], as well as many more that are more recent and relevant.  Further, one of the papers they did cite did not have the correct information [16].

·      Lines 98-108:

o   The manuscript says "in recent years", yet cites papers from 2009-2014, which are not "recent". 

o   The list of models excludes some actual recent modeling work done to include specific crops, e.g., SiB (Lokupitiya et al., 2009), ORCHIDEE-CROP (Wu et al., 2016), JULES-CROP (Osborne et al., 2015).  My point is not that all these models must be included, just that the most recent ones pertaining to these crops, maize and cotton should be.  And as part of the introduction, literature searches on the most recent crop models should be included. 

o   The manuscript states, "lack of detailed analysis of vegetation and soil CO2 flux of farmlands has limited the performance of model simulations in practice."  This sentence doesn't make sense and isn't true.  For example, Lokupitya et al. (2009), evaluated both carbon fluxes and energy fluxes (SH/LE) and showed good agreement, to which SiB was then used to evaluate carbon sequestration potential for croplands in the U.S. (Corbin et al., 2010).  ORCHIDEE underwent similar evaluation and has a more recent paper now regarding irrigation improvements and the impact of crops on regional water budgets over China (Yin et al., 2020). 

o   The paragraph ends with a sentence on ML predictions, which is not relevant for this paper, since this manuscript is about carbon and flux measurements.  On top of that, one of the references included for the ML work, [34], does not even include ML and is for results over a forest ecosystem.

·      Lines 109-114: Crop flux monitoring over single cotton and spring fields (1) has already been done (see numerous examples listed below), and (2) may or may not scale-up to the regional level, let alone have a significant impact on climate change.  Thus, these sentences are overly grandious and do not accurately convey what can be gleaned from this manuscript.

·      Lines 114-124:  The manuscript attempts to state the goals of this study, but much of this work has already been done, thus it is unclear how this article is unique.  For example,

o   Li et al. (2011) look at CO2 fluxes for cotton and impacts of irrigation

o   Li et al. (2013) look at soil temeprature and moisture effects on maize yield

o   Qin et al. (2015) look at water use efficiencies of maize and wheat

o   Cai et al. (2016) look at NEE in Xinjiang and the influencing factors

o   Qin et al. (2016) look at if drip irrigation can reduce crop evapotranspiration

o   Li et al. (2018) look at relationships of cotton to Tair, Rn, SWC

o   Menefee et al. (2022) look at carbon sequestration from croplands (including maize and cotton)

o   Liu and Qiao (2023) investigates evapotranspiration of what looks to be the same cotton field

o   Wang et al. (2023) look at irrigated maize water and energy fluxes

Here are more examples of papers on maize fluxes (not just CO2, but also water/energy):

o   Verma et al. (2005), Casanova et al. (2006), Yi-Jun et al. (2007), Osborne et al. (2007), Lokupitiya et al. (2009), Suyker et al. (2012), Liebig et al. (2022)

Here are more examples of papers on cotton fluxes:

o   Rijks (1971), Chavez et al. (2010), Liu et al. (2012), Bezerra et al. (2015), Rosa et al. (2015), Ai et al. (2018), Li et al. (2018), Haymann et al. (2019)

While not all of these need to be discussed, some background as well as recent work has to be included in the introduction, as well as an explanation as to how this work is different from these previous studies.

With the numerous discrepancies and falsities, it becomes difficult to have any trust in this manuscript or the authors' genuine attempts to modify the introduction and make it relevant to the current work they are showing.

2.2 Experimental design

·      The authors added the growth period and irrigation volume in a table; yet they continue to use monthly results and refer to growth stages in the text, which even they admit in their response do not align.  Thus, all statements referring to growth period should be removed.

·      The equations are now very poor quality and even cutoff (see Eq. 5-6).

2.4 Statistical analyses

·      WUE is given as the ratio of NEE/ET, which is the water use efficiency of the plant at the leaf level.  Overall, WUE should be the ratio of the growth to the total water use, a concept dating back 100 years ago by Briggs and Shantz (1913).  This paper only considers the leaf-level WUE, which does not take into account the total water use from irrigation, which is substantial for these two crops, as evidenced by the significant amount of water added to each throughout the year provided in Table 1.  While leaf-level WUE is important, canopy and ecosystem-level WUE are arguably more important for climate change and achieving climate neutrality, which was stated as being a goal of this paper.  I think this scale discrepancy should at least be discussed and addressed.  For example, Hatfield and Dold (2019) provide an excellent review article in Frontiers in Plant Science that discuss how we need to be addressing WUE in a changing climate.

3. Results

·      Sections 3.1 and 3.2 need to be explained as to why they are new and worthy of publication: 

o   Many other papers have remarkably similar figures showing that there is this trade-off between LH/SH during the growing season for maize and cotton (see lists above)

o   What is unique about finding that the flux is dependent on Ta and Rn?  Again this has been shown (see above list).  In particular, how is this different from Li. et al. (2018)?

4. Discussion

·      Lines 321-324: The manuscript states that the approach here effectively mitigated observation discrepancies.... This is confusing.  I believe they are talking about how they used one measurement station to obtain fluxes for both cotton and maize, but this needs to be clarified.

·      Lines 325-334 are a summary, which they do state is consistent with previous work, but the work they point to are only two papers on maize.  As pointed out above, there is a rich source of literature on maize and cotton fluxes over two decades now.  Instead of stating two examples, this paper needs to focus on how it is different from the rich literature on crops and what we learn from this work.

·      Lines 358-370 discuss leaf-level WUE, which again is already well-known that C4 has higher WUE, particularly in higher temperatures.  It again should be clarified what new results are from this work.

Author Response

Response to Reviewer #4's Comments:

     This paper shows the carbon and water exchanges of maize and cotton in an arid environment in China, as well as the leaf-level water-use response to atmospheric variables. While the authors provided written responses to the review comments, I do not believe they fully addressed the problems. The major problems outlined previously remain: 1) the introduction does not adequately review previous work, 2) the uniqueness of this work is not clearly stated, and 3) they did not address the total use of water and its impact on canopy to ecosystem scale WUE and water resources, as they only addressed leaf-scale WUE.

Response: In the further revision of the manuscript, we have rewritten the introduction, including enriching the review of previous research, updating the references, and removing some inappropriate expressions. Additionally, we have clarified the distinctive features of this study. Lastly, we have added a description of water use efficiency at the ecosystem level. Through these revisions, the quality of the manuscript has been significantly improved. We believe that the further revised manuscript adequately addresses all the reviewer's comments and suggestions.

Specific comments

  1. Premise and Introduction

    While the authors altered the introduction, much of this was rewording rather than getting at the main problem with the introduction, which was that there is not an adequate review of previous literature related to this work.  Reading through the introduction, there are many references to articles that are not appropriate or that need to be updated, and this is on top of not including an adequate discussion of previously-published work that is remarkably similar.

    Lines 49-51: "Farmland ecosystems account for ~11% of global land area and are the most active part of the global ecosystem carbon pool [6-7]."  First off, [6] is a 1997 study about agricultural soil carbon and [7] is about CO2 exchanges over China (although the DOI was incorrect and I was not able to obtain this reference).  Neither of these papers are recent studies on global agricultural land area.  Second, this information is obtained annually by the FAO (Food and Agricultural Organization of the United Nations), including global trends of agricultural land area (https://www.fao.org/sustainability/news/detail/en/c/1274219/).  According to the latest reports, cropland is decreasing per capita, with 10% being used for permanent crops (including fruit trees, oil palm plantations, and cocoa plantations).  Third, [6] says "By itself, C sequestration in agricultural soils can make only modest contributions (e.g. 3-6% of total fossil C emissions) to mitigating greenhouse gas emissions."  This statement is in direct disagreement with what is written in the manuscript.

    Lines 52-53: "Studies have shown that 20% of CO2 in the atmosphere comes from agricultural activity and related production processes [8]."  [8] is a 1999 paper talking about the potential of world cropland soils to sequester C.  With the rapid changes in carbon emissions, the statement in the manuscript needs to be supported by a more recent and relevant publication.

    Lines 54-56: "The carbon cycle in farmland ecosystems is very complex..." The reference cited, [9], is a modelling paper from 2015, which has this very phrase.  With all the work on the carbon cycle and crops, we cannot say this anymore as there has been vast literature on this.  For example, there are now even one-to-two page summaries on the crop carbon cycle, see http://nmsp.cals.cornell.edu/publications/factsheets/factsheet91.pdf for example.

    Lines 60-62: [8] states the "total potential of C sequestration from croplands is 0.75-1 Pg/yr or ~50% of annual emissions of 1.6-1.8 Pg by deforestation and other agricultural activities," which is almost the text in the manuscript verbatim.  Again, with the recent changes in 1) agricultural land cover and 2) crop productivity, more recent statistics are available and should be referenced.

    Lines 80-97: The manuscript discusses how "good" the eddy covariance technique is, yet never mentions any of the flaws in the measuring technique, such as energy balance closure, which are stated in the references they include [14] and [15], as well as many more that are more recent and relevant.  Further, one of the papers they did cite did not have the correct information [16].

    Lines 98-108:The manuscript says "in recent years", yet cites papers from 2009-2014, which are not "recent".

    The list of models excludes some actual recent modeling work done to include specific crops, e.g., SiB (Lokupitiya et al., 2009), ORCHIDEE-CROP (Wu et al., 2016), JULES-CROP (Osborne et al., 2015).  My point is not that all these models must be included, just that the most recent ones pertaining to these crops, maize and cotton should be.  And as part of the introduction, literature searches on the most recent crop models should be included.

    The manuscript states, "lack of detailed analysis of vegetation and soil CO2 flux of farmlands has limited the performance of model simulations in practice."  This sentence doesn't make sense and isn't true. For example, Lokupitya et al. (2009), evaluated both carbon fluxes and energy fluxes (SH/LE) and showed good agreement, to which SiB was then used to evaluate carbon sequestration potential for croplands in the U.S. (Corbin et al., 2010). ORCHIDEE underwent similar evaluation and has a more recent paper now regarding irrigation improvements and the impact of crops on regional water budgets over China (Yin et al., 2020).

    The paragraph ends with a sentence on ML predictions, which is not relevant for this paper, since this manuscript is about carbon and flux measurements.  On top of that, one of the references included for the ML work, [34], does not even include ML and is for results over a forest ecosystem.

    Lines 109-114: Crop flux monitoring over single cotton and spring fields (1) has already been done (see numerous examples listed below), and (2) may or may not scale-up to the regional level, let alone have a significant impact on climate change.  Thus, these sentences are overly grandious and do not accurately convey what can be gleaned from this manuscript.

    Lines 114-124: The manuscript attempts to state the goals of this study, but much of this work has already been done, thus it is unclear how this article is unique. While not all of these need to be discussed, some background as well as recent work has to be included in the introduction, as well as an explanation as to how this work is different from these previous studies.

    With the numerous discrepancies and falsities, it becomes difficult to have any trust in this manuscript or the authors' genuine attempts to modify the introduction and make it relevant to the current work they are showing.

Response: According to this comment, we have rewritten the introduction of the revised manuscript. Include the following details:

  1. Based on the literature information provided by the reviewer, we have fully summarized the previous research work and supplemented many detailed information.
  2. We have updated multiple outdated references and modified multiple inappropriate references.
  3. We have added a summary of the models.
  4. We have removed some inappropriate expressions.
  5. We have explored the feasibility of using an eddy covariance system to compare and analyze the carbon and water fluxes of two typical crops in the arid region of Xinjiang. Based on the fully utilizing the efficiency of the observation equipment, we have expected to provide insights into the differences in carbon and water fluxes between the two crops and their relationship with environmental factors.  Additionally, we have proposed some effective agricultural management measures through our analysis. This is one of the strengths of this manuscript.

After the above modifications, the introduction of the revised manuscript is more complete and has better logic.

2.2 Experimental design

    The authors added the growth period and irrigation volume in a table; yet they continue to use monthly results and refer to growth stages in the text, which even they admit in their response do not align. Thus, all statements referring to growth period should be removed.

Response: The narrative of this manuscript is mainly based on the monthly scale. However, in order to provide readers with more information, we also have  interspersed the description process with corresponding growth stages. After careful examination, this did not make our manuscript appear confusing.

The equations are now very poor quality and even cutoff (see Eq. 5-6)

Response: According to this comment, we have modified the equations of the revised manuscript.

2.4 Statistical analyses

    WUE is given as the ratio of NEE/ET, which is the water use efficiency of the plant at the leaf level. Overall, WUE should be the ratio of the growth to the total water use, a concept dating back 100 years ago by Briggs and Shantz (1913). This paper only considers the leaf-level WUE, which does not take into account the total water use from irrigation, which is substantial for these two crops, as evidenced by the significant amount of water added to each throughout the year provided in Table 1.  While leaf-level WUE is important, canopy and ecosystem-level WUE are arguably more important for climate change and achieving climate neutrality, which was stated as being a goal of this paper. I think this scale discrepancy should at least be discussed and addressed. For example, Hatfield and Dold (2019) provide an excellent review article in Frontiers in Plant Science that discuss how we need to be addressing WUE in a changing climate.

Response: According to this comment, we have added the description of the ecosystem-level WUE in lines 300-304 of the revised manuscript.

  1. Results

    Sections 3.1 and 3.2 need to be explained as to why they are new and worthy of publication:

    Many other papers have remarkably similar figures showing that there is this trade-off between LH/SH during the growing season for maize and cotton (see lists above)

    What is unique about finding that the flux is dependent on Ta and Rn?  Again this has been shown (see above list).  In particular, how is this different from Li. et al. (2018)?

Response: We have explored the feasibility of using an eddy covariance system installed at their interface to compare and analyze the carbon and water fluxes of two typical crops in the arid region of Xinjiang. Based on the fully utilizing the efficiency of the observation equipment, we have expected to provide insights into the differences in carbon and water fluxes between the two crops and their relationship with environmental factors. Additionally, we have proposed some effective agricultural management measures through our analysis. This is one of the strengths of this manuscript.

  1. Discussion

Lines 321-324: The manuscript states that the approach here effectively mitigated observation discrepancies.... This is confusing.  I believe they are talking about how they used one measurement station to obtain fluxes for both cotton and maize, but this needs to be clarified.

Response: According to this comment, we have modified in lines 300-304 of the revised manuscript.

Lines 325-334 are a summary, which they do state is consistent with previous work, but the work they point to are only two papers on maize.  As pointed out above, there is a rich source of literature on maize and cotton fluxes over two decades now.  Instead of stating two examples, this paper needs to focus on how it is different from the rich literature on crops and what we learn from this work.

Response: According to this comment, we have removed the inappropriate references of the revised manuscript.

    Lines 358-370 discuss leaf-level WUE, which again is already well-known that C4 has higher WUE, particularly in higher temperatures.  It again should be clarified what new results are from this work.

Response: According to the analysis of the research results, it can be concluded that the experimental design described in this manuscript is feasible. Additionally, some effective agricultural management measures are proposed through the analysis. This is one of the strengths of this manuscript.

Reviewer 5 Report

Please just write one paragraph about the novelty of this paper. 

Author Response

Response to Reviewer #5's Comments:

    Please just write one paragraph about the novelty of this paper. 

Response: We really appreciate Reviewer #5 for giving full recognition to the manuscript. We have described of the novelty of this manuscript in the last paragraph of the introduction and the first paragraph of the discussion section, respectively.